# A High-Throughput Screening of a Natural Products Library for Mitochondria Modulators

**DOI:** 10.3390/biom14040440

**Published:** 2024-04-04

**Authors:** Emmanuel Makinde, Linlin Ma, George D. Mellick, Yunjiang Feng

**Affiliations:** 1Griffith Institute for Drug Discovery, Griffith University, Brisbane, QLD 4111, Australia; emmanuel.makinde@griffithuni.edu.au (E.M.); linlin.ma@griffith.edu.au (L.M.); g.mellick@griffith.edu.au (G.D.M.); 2School of Environment and Science, Griffith University, Brisbane, QLD 4111, Australia

**Keywords:** mitochondria modulators, high-throughput screening, natural products

## Abstract

Mitochondria, the energy hubs of the cell, are progressively becoming attractive targets in the search for potent therapeutics against neurodegenerative diseases. The pivotal role of mitochondrial dysfunction in the pathogenesis of various diseases, including Parkinson’s disease (PD), underscores the urgency of discovering novel therapeutic strategies. Given the limitations associated with available treatments for mitochondrial dysfunction-associated diseases, the search for new potent alternatives has become imperative. In this report, we embarked on an extensive screening of 4224 fractions from 384 Australian marine organisms and plant samples to identify natural products with protective effects on mitochondria. Our initial screening using PD patient-sourced olfactory neurosphere-derived (hONS) cells with rotenone as a mitochondria stressor resulted in 108 promising fractions from 11 different biota. To further assess the potency and efficacy of these hits, the 11 biotas were subjected to a subsequent round of screening on human neuroblastoma (SH-SY5Y) cells, using 6-hydroxydopamine to induce mitochondrial stress, complemented by a mitochondrial membrane potential assay. This rigorous process yielded 35 active fractions from eight biotas. Advanced analysis using an orbit trap mass spectrophotometer facilitated the identification of the molecular constituents of the most active fraction from each of the eight biotas. This meticulous approach led to the discovery of 57 unique compounds, among which 12 were previously recognized for their mitoprotective effects. Our findings highlight the vast potential of natural products derived from Australian marine organisms and plants in the quest for innovative treatments targeting mitochondrial dysfunction in neurodegenerative diseases.

## 1. Introduction

The mitochondrion, often referred to as the cell’s “powerhouse” is pivotal for the functioning of eukaryotic cells, as it is responsible for most of the chemical energy supply needed to fuel cellular activities. This energy is mainly produced through oxidative phosphorylation (OXPHOS), a process that generates chemical energy stored as adenosine triphosphate (ATP), which is used to power complex biochemical processes in the cell [1,2]. In addition to their central role as the site of chemical energy generation, mitochondria are also crucial in regulating healthy cellular apoptosis, calcium homeostasis, biosynthesis of lipids and amino acids, and generating reactive oxygen species (ROS).

Neurons, with their high energy demands, contain hundreds of thousands of mitochondria, which are responsible for meeting most of their ATP needs for the functioning of the CNS [3,4,5]. The quality of these mitochondria is critical, as they must be highly functional to support the complex activities of neurons in the CNS [3]. Dysfunction in mitochondria, particularly in the OXPHOS system, has been linked to various diseases, notably neurodegenerative disorders like Parkinson’s disease (PD), Huntington’s disease, and Alzheimer’s disease [2,3,6,7,8,9]. The mechanisms behind these links are multifaceted.

Firstly, diminished ATP production due to impaired mitochondrial function plays a significant role in the energy deficits observed in neurons affected by these diseases. Such deficits can compromise neuronal function and survival, potentially leading to cell death [3,5,10]. Secondly, about 90% of ROS are generated as by-products of the OXPHOS process [1,6,11,12]. Although ROS serve as signaling molecules under normal conditions, their overproduction or the failure of antioxidant defenses can induce oxidative stress, harming DNA, proteins, and lipids. This oxidative damage, a common feature of neurodegenerative diseases, likely drives further neuronal damage [3,5]. Thirdly, mitochondria play a key role in regulating intracellular calcium levels, which are crucial for various cellular processes, including neurotransmitter release, synaptic plasticity, and cell survival. Mitochondrial dysfunction can lead to dysregulated calcium homeostasis, exacerbating neuronal injury and death [1,6]. Fourthly, impaired mitophagy, a specific form of autophagy that removes damaged mitochondria from the cell, has been linked to neurodegenerative diseases such as PD [13,14]. Mutations in genes like PINK1 and Parkin, which are involved in mitophagy, can cause genetic PD, underscoring the importance of mitochondrial quality control in neurodegeneration [15,16]. Lastly, mitochondria play a crucial role in the intrinsic apoptosis pathway by releasing pro-apoptotic factors such as cytochrome C. Dysregulation of apoptotic signaling pathways due to mitochondrial dysfunction can trigger inappropriate neuronal cell death, contributing to neurodegeneration [12,17,18].

Recognizing the pivotal role of mitochondrial function in various diseases, extensive research has focused on identifying compounds that can modify mitochondrial functions. Compounds such as berberine, resveratrol, and epigallocatechin-3-gallate are known to trigger mitochondrial biogenesis and influence mitochondrial dynamics by promoting both fusion and fission [19,20,21,22,23,24,25]. Curcumin and its derivatives have been demonstrated to regulate mitochondrial dynamics to remedy dysfunction, and flavonoids like quercetin have shown potential in ameliorating memory impairment through mitochondrial regulation [26,27,28,29,30,31]. Additionally, compounds such as polydatin and acacetin have been found to induce mitophagy, enhancing mitochondrial function and offering protective effects in disease models [32,33]. Although these products show promise in enhancing mitochondrial function and offering disease model protection, the variability in the effectiveness of these compounds across different cell types and disease models highlights the need for more targeted approaches in their application and calls for novel, safe, and more potent mitochondrial modifiers.

Throughout history, natural products have served as the foundation of modern drug discovery. Notably, about half of FDA-approved drugs are either unmodified natural products or their synthetic derivatives [34,35]. Interest in natural product drug discovery seems to have waned over time; however, there has been a resurgence of interest in and commitment to natural product drug discovery in recent years [36,37]. This, in part, is due to technological advancement that has made it possible to screen thousands to hundreds of thousands of molecules against disease targets in a very short timeframe [36,37,38].

In our search for mitochondrial modulators with protective effects, we have established high-throughput screening assays and tested fractions sourced from a wide variety of Australian plants and marine sponges. These fractions have been obtained using a customized lead-like fractionation protocol developed in our laboratory [36]. To target lead-like natural products more efficiently, we diverged from the conventional approach of testing crude extracts. Instead, we directly examined 4224 fractions from 384 biota samples employing the CyQuant assay, a highly sensitive, quick, and robust cell viability assay [39]. Twenty fractions, representing 11 biota samples, were identified to protect the cells from rotenone, a mitochondrial complex I inhibitor [40]. Subsequently, the fractions from 11 active biota were further evaluated for their mitochondria modulatory activities using an MTT assay targeting mitochondrial nicotinamide adenine dinucleotide phosphate (NADPH)-dependent dehydrogenases [41,42], as well as a mitochondrial membrane potential (MMP) assay for protection against neurotoxins. Finally, the most active fractions were subjected to LC-Orbitrap MS analysis to identify natural products that can defend the mitochondria from toxic assaults (Figure 1).

A common approach utilized to search for mitoprotective metabolites is to test compounds largely based on previously reported antioxidant or neuroprotective activity [43,44,45]. This work diverges from that by screening a large library of natural products to boost the chances of finding novel mitoprotective compounds. Overall, we have successfully established and implemented a robust process for the identification of mitoprotective compounds from natural products using a stepwise combination of three different assays, two cell lines, and two mitochondrial toxins. We have also described how each assay seamlessly dovetails with the next, leading to the identification of 57 metabolites, including 45 new mitoprotective compounds.

## 2. Materials and Methods

### 2.1. Ethics Statement

This work uses patient-derived cells which were collected under the ethical approval of the Griffith University ethics committee. Human olfactory neurosphere-derived cells were derived from nasal biopsies following approved ethical standards, as previously reported by Cook et al. [46]. The human neuroblastoma cell line SH-SY5Y was obtained from the ATCC (CRL-2266), Manassas, VA, USA.

### 2.2. Cell Culture

Human olfactory neurosphere-derived (hONS) cells were obtained from biopsies of the olfactory mucosa of PD patients and healthy controls [47,48]. The hONS cell line was cultured in DMEM/F-12 medium (Gibco, Invitrogen, Waltham, MA, USA) supplemented with 10% fetal bovine serum (Gibco, Invitrogen) and incubated at 37 °C with 5% CO_2_. 

The SH-SY5Y cells were cultured in DMEM/F-12 (Sigma Aldrich, St. Louis, MI, USA) medium supplemented with 10% fetal calf serum, 5% non-essential amino acids, and 5% glutamax (Gibco, Invitrogen) and incubated at 37 °C with 5% CO_2_. 

### 2.3. Plant Material

Biota samples were obtained from NatureBank, Vancouver, BC, Canada, a unique drug discovery platform that focuses on extracts and fractions of a diverse range of natural products sourced from Australian plants, fungi, and marine organisms [49]. The 384 biota samples screened in this project were Australian plants and marine sponges sourced from Queensland and were selected based on availability at the request time. 

### 2.4. Extraction and Fractionation

The 384 biotas obtained from NatureBank were extracted using a modified lead-like extraction protocol previously established by NatureBank [36,50]. Briefly, plant material (600 mg) was washed in a solid-phase extraction cartridge with 4 mL of hexane to remove fatty components, followed by extraction with 4 mL of dichloromethane and methanol, successively. The hexane phase was discarded, while the dichloromethane and methanol extracts were combined and dried. The dried extracts were reconstituted in methanol and passed through polyamide gel to remove tannins, and the extracts were dried and stored. Marine sponges were extracted in a similar way with 4 mL dichloromethane/methanol (80:20 *v*/*v*) followed by 4 mL methanol.

Extracts were fractionated using the lead-like fractionation protocol developed by NatureBank [50]. Briefly, the lead-like extracts dissolved in DMSO were subjected to HPLC separations using a Phenomenex Onyx Monolithic C_18_ column (4.6 mm × 100 mm) with a gradient solvent system of MeOH:H_2_O (0.1% TFA), as shown in Table 1. Fractions were collected every 60 s over 11 min. A total of 4224 fractions were collected for testing.

### 2.5. Cell Viability Assays

CyQuant assay

A CyQUANT Cell Proliferation Assay Kit (Life Technologies, Carlsbad, CA, USA) was used to evaluate the cellular response of hONS cells to rotenone, as described by Murtaza et al. [47]. Cells were seeded at 625 cells per well in a 384-well plate and treated with 200 nm rotenone for 96 h. The fluorescence intensity of each sample was measured using the Synergy2 plate reader (Biotek Instruments, Winooski, VT, USA) set with an excitation of 485 nm and emission detection at 530 nm.

MTT assay

To enhance cell adhesion, all plates were pre-treated with 0.01% *w*/*v* poly-D-lysine (Sigma Aldrich, St. Louis, MI, USA) at least 3 h before the assay. Cells were seeded at an optimized density of 5 × 10^4^ cells/well in a 96-well plate, leaving 3 empty wells as controls. The MTT assay was performed as previously described [51], the cells were incubated at 37 °C for 24 h, and then treated with 100 µg/mL of DMSO-solved fractions for 2 h at a final DMSO concentration of 0.5%. After pretreatment with fractions, cells were subjected to 6-OHDA challenge at the working concentration of 60 µM in FBS-free media, followed by incubation for another 24 h. To quantify cell viability based on mitochondrial function, the cells were incubated with 0.5 mg/mL MTT and incubated for 4 h at 37 °C. To dissolve insoluble formazan crystals produced from reducing MTT by the cells, 80 µL of 20% SDS was added, and plates were wrapped with foil and placed on an orbital shaker at 100 rpm for 4 h at room temperature. The absorbance of solubilized formazan products was then measured at 570 nm. 

### 2.6. Mitochondrial Membrane Potential (MMP) Assay

A Codex Homogeneous Mitochondrial Membrane Potential Assay Kit (Codex BioSolutions, Inc., Gaithersburg, MD, USA) was used for the MMP assay; the assay was performed in line with the manufacturer’s protocol and as described by Sakamuru et al. [52] with some modifications. To determine the appropriate toxin dose, cells were seeded at an optimized density of 5 × 10^4^ cells/well in a 96-well plate. After 24 h, culture media were replaced with 100 µL media containing 6-OHDA concentrations ranging from 1.56 µM to 200 µM and incubated for 6 h at 37 °C. When fractions were tested, fractions solved in DMSO were added to the wells at 100 µg/mL 30 min after the addition of 6-OHDA at the optimized concentration. After incubation, the cells were loaded with the mitochondrial membrane potential indicator (m-MPI) and incubated at 37 °C for 30 min. The plate was subsequently washed once with 1X m-MPI assay buffer, after which 80 µL of 1X m-MPI assay buffer was added to each well for reading of the plate with a Spectramax iD5 Multi-Mode Microplate reader. The mitochondrial membrane potential was quantified by the ratio between the J-aggregate form of the mitochondrial MPI indicator (m-MPI) with green fluorescence (485 nm excitation and 535 nm emission) and the monomer form of the m-MPI with red fluorescence (540 nm excitation and 590 nm emission). 

### 2.7. Untargeted Phytochemical Characterization of Active Fractions by HRMS

Acquisition of the phytochemical profile of fractions was carried out on a Vanquish™ Flex UHPLC system (Thermo Fisher Scientific, Waltham, MA, USA) connected to an Orbitrap Exploris 120 mass spectrometer (Thermo Scientific, Waltham, MA, USA). Separation of fractions was achieved on a Phenomenex Luna C18 column, (2.1 mm × 100 mm, 1.7 μm) using a mobile phase of A: 0.1% formic acid in water and B: methanol at a flow rate of 0.6 mL/min. The gradient program was as follows: 0 min, 10% B; 5 min, 10% B; 15 min, 100% B; 20 min, 10% B. The injection volume was 5 µL, and the column temperature was set at 35 °C.

Mass spectrometry data were recorded on an Orbitrap Exploris 140 mass spectrometer equipped with a heated ESI source and operated in the positive-ion mode with the following settings: ion spray voltage: 2.5 kV, sheath gas: 5.08 L min^−1^, auxiliary gas: 9.37 L min^−1^, ion transfer tube temperature: 320 °C, vaporizer temperature: 350 °C, scan range (*m*/*z*): 150–2000, and collision-energy voltage: 35 V. The full scan was operated at a mass resolution of 60,000 and MS^2^ scan at 15,000. Data were acquired using Thermo Xcalibur software and analyzed with Compound Discoverer 3.3.

### 2.8. Data Analysis

Data analysis was performed using Graph Pad Prism version 10.1 for Microsoft windows (GraphPad Software, San Diego, CA, USA) using two-way analysis of variance, followed by Dunnett’s multiple comparison test. Statistical significance was defined as * *p* < 0.05 and ** *p* < 0.01. All determinations were performed in triplicate (at least), and results are presented as means ± SDs.

## 3. Results and Discussion

### 3.1. Extraction and Fractionation 

A total of 384 biota were randomly chosen from NatureBank, and the extracts and fractions were obtained using a modified lead-like extraction and fractionation protocol previously established by NatureBank [36,50]. This protocol has been optimized to prioritize molecules with drug-like physiochemical properties by frontloading extracts and subsequent fractions while excluding molecules that lack drug-like or lead-like characteristics [36,50]. Generally, the lead-like extraction and fractionation protocol retains components with a log P < 5 [36,50]. Extracts were then fractionated using HPLC, with fractions collected at 60 s intervals for 11 min (Figure 2), resulting in 4224 fractions for screening.

### 3.2. Effect of Fractions on Cell Proliferation Using High-Throughput CyQuant Assay

The effect of fractions used in this study on cell proliferation was evaluated by exposing PD-sourced hONS cells to rotenone in the presence of test fractions. There is a well-established link between PD and mitochondrial dysfunction, as extensive research evidence has linked PD to features of mitochondrial dysfunction, such as oxidative stress, poor calcium homeostasis, disruption to the MMP, abnormal mitochondrial morphology, disruptions to the OXPHOS process as a result of reduced complex enzyme activity, imbalanced apoptosis, impaired mitophagy, and mitochondrial dynamics [3,14,53,54,55,56]. 

PD-patient derived hONS cells have been reported to show disease-specific alterations such as mitochondrial dysfunction and oxidative damage even in the absence of mitochondrial toxins [48]. Hence, it was possible to aggravate mitochondrial dysfunction in these cells and cause apoptosis using very low doses of neurotoxins, such as the 200 nm rotenone applied in this study. Rotenone is a well-known neurotoxin that induces mitochondrial dysfunction in cellular and animal disease models [40,47,57]. Its primary mechanism of action involves causing mitochondrial dysfunction by inhibiting mitochondrial complex I, an enzyme critical to the oxidative phosphorylation process [40]. Numerous studies have shown that the inhibition of mitochondrial complex I activity is accompanied by deleterious effects, such as impaired mitochondrial biogenesis and dynamics, oxidative stress, decreased ATP production, and apoptosis [40,57]. To identify modulators within the 4224 selected fractions affecting rotenone-induced toxicity in hONS cells, we first employed the CyQuant assay, a fluorescence-based method for quantifying and assessing cell proliferation and cytotoxicity [39]. We calculated Z-scores for all vehicle control-treated samples and set a threshold of a Z-score in the range of −2.5–2.5. The CyQuant values for all extract-treated samples were normalized to control-treated samples, and a relative Z-score was calculated based on the means and standard deviations of all control-treated samples. Figure 3 shows the relative Z-scores for all samples together with the threshold values as indicated by the two red lines. There were 108 fractions with relative Z-scores outside the threshold range; these constituted the initial hits. The 108 fractions were subsequently subjected to a confirmatory round (in triplicate) of the assay using the same parameters as the first. This resulted in 20 hit fractions with relative Z-scores ranging from −2.5 to 2.5. The 20 fractions were from 11 biotas (Table 2), which included eight Australian plants and three marine sponges (Figure 4).

### 3.3. Evaluating the Mitoprotective Effects of Hit Fractions by MTT Assay

To further validate the hits from the high-throughput CyQuant assay and assess their efficacy and potency, different cell models and assays were employed. Mitochondria exhibit tissue-specific properties and are known to show differences across tissues and cell lines [58]. Furthermore, PD patient-sourced hONS cells are primary cells, presenting diverse genetic and morphological characteristics among individual PD patients, which further complicates the validation process [48]. Given these considerations, we opted to confirm our findings from the CyQuant high-throughput screening using the human neuroblastoma cell line SH-SY5Y. SH-SY5Y cells resemble human dopaminergic neurons in many aspects and are widely used as model cells for studying mitochondrial dysfunction [59]. A recent review highlighted that approximately 41% of studies focusing on mitochondrial dysfunction used the SH-SY5Y cell line [1]. This is unsurprising, given that neuronal cells have high energy demands. Mitochondria play a crucial role in their health and functionality, catering to their high metabolic needs [1,2,3]. To establish a robust assay, we optimized conditions such as cell seeding density, the choice of neurotoxin and toxin concentration, and the DMSO tolerability of the cells. An optimized seeding density of 5.0 × 10^4^ cells/well in a 96-well plate was established for SH-SY5Y cells, and three mitochondrial complex I inhibitory neurotoxins, namely, 6-hydroxydopamine (6-OHDA) [60,61], rotenone [40], and 1-methyl-4-phenylpyridinium (MPP^+^), were tested [59]. DMSO was used as a carrier control. 

As shown in Figure 5, rotenone showed a relatively low apparent IC_50_ of 0.59 µM. However, it presented minimal efficacy of ~30% due to a significant solubility challenge, as it forms precipitates in culture media. MPP^+^ has no solubility issues. However, it exhibited very low potency towards the SH-SY5Y cells. In contrast, 6-OHDA showed almost 100% efficacy and a moderate potency. Therefore, 6-OHDA was selected as the most suitable toxin to induce mitochondrial dysfunction, and a working concentration of 60 µM was used to maintain a cell viability range of 50–70%, allowing for scalability in the following large-scale experiments. 

SH-SY5Y cells can tolerate a DMSO concentration of up to 1% *v*/*v* in culture media, maintaining at least 91% cell viability. To keep the DMSO concentration minimal without affecting the solubility of the fractions, the DMSO concentration was kept at 0.5% in the media containing tested fractions in the following studies.

For all 11 biotas, fractions 1–7 were tested for their protective effect on the mitochondria at concentrations of 100, 50, and 25 µg/mL, using the optimized MTT assay conditions (Figure 6). We excluded fractions 8–11 due to their minimal mass, which rendered them unsuitable for functional tests. The controls encompassed untreated cells, cells treated with DMSO, cells treated with 6-OHDA, and cells treated with 6-OHDA + DMSO. Treated groups were cells incubated with fractions in the presence of 6-OHDA and DMSO. Any fraction with statistically significant cell viability above that of cells treated with 6-OHDA and DMSO, which was 38.98%, was considered protective. Due to the number of data, only fractions showing significant protection against 6-OHDA-induced toxicity are shown in Figure 6. However, the full data sets for all 11 biotas, including inactive fractions, have been included in the Appendix A. A total of 77 fractions were tested, and 40 (51.94%) fractions showed statistically significant protection (*p* < 0.05) against 6-OHDA-induced toxicity at a treatment concentration of 100 µg/mL (Figure 6A). Furthermore, 28 (36.36%) and 31 (40.26%) fractions showed statistically significant protection at lower concentrations of 50 and 25 µg/mL, respectively (Figure 6B,C).

To a very large extent, the data from the MTT assay corroborate the results from the CyQuant assay, with 18 of the initial 20 hits from the CyQuant assay returning as positive despite the different cell model and assay. This, essentially, is a strong indicator that these fractions potentially contain protective mitochondrial modulators.

It is also noteworthy that fraction F7 of *Ternstroemia* sp. (Appendix A), F1–F7 of *Ilex* sp. (Appendix A), and F1–F7 of *Furcraea* sp. (Appendix A) showed significant toxicity to the mitochondria at both 50 and 25 µg/mL, resulting in lower cell viability when compared to the control group. These results suggest that these fractions could potentially contain mitochondrial modulators with toxic effects on the mitochondria. This is remarkable, and compounds responsible for this effect are potentially relevant in the treatment of diseases like cancer, with the mitochondrion strongly emerging as a target for cancer therapy, as there is increasing research evidence showing that inducing mitochondrial toxicity by targeting OXPHOS could be a potent way to destroy cancer cells [62,63,64].

### 3.4. Evaluating the Mitoprotective Effects of Fractions by Mitochondrial Membrane Potential (MMP) Assay 

To further ascertain that the active fractions identified from the MTT assay were indeed acting on mitochondria, it was essential to use an assay with greater mitochondrial specificity. Hence, the active fractions were further tested using the mitochondrial membrane potential (MMP) assay.

ATP is generated in the mitochondria through the electron transport chain by creating an electrochemical gradient through a series of redox reactions [52,65]. This electrochemical gradient generates MMP, which is a direct measure of ATP production ability and a key metric for evaluating mitochondrial function and overall cellular health [65,66]. A decrease in MMP is one of the main features of mitochondrial dysfunction, and this decrease has been linked to apoptosis [52,66,67].

MMP was evaluated using the water-soluble mitochondrial membrane potential indicator (m-MPI), a dye that aggregates in healthy mitochondria as red fluorescent monomers emitting light at 590 nm [52]. When the MMP is depolarized, the m-MPI dye transitions to green fluorescent monomers with emission at 535 nm. The ratio between these two fluorescence values can then serve as a measure of MMP [52,67]. A total of 34 fractions with at least 60% efficacy at 100 µg/mL from the MTT assay were subjected to the MMP assay. We found that 6-OHDA induced mitochondrial dysfunction in SH-SY5Y cells by reducing MMP, with a mere 1.56 µM concentration decreasing MMP by more than 56% (Figure 7A). All the tested fractions, except for fraction F7 of *Dendrilla* sp. (11.42%), restored MMP to levels ranging between 36% and 134% when compared to cells treated with 6-OHDA and DMSO, where the level was 20.56% (Figure 7B). Notably, fraction F2 of *Balanophora* sp. fully restored MMP, followed by fractions F3 of *Dendrilla* sp. and F6 of *Aaptos* sp., which restored MMP to levels above 80%.

These findings corroborate the data from the MTT assay and further confirm that the selected fractions potentially contain components capable of rescuing mitochondria from toxic attacks. 

### 3.5. Chemical Constituents of Selected Active Fractions

In Table 3, we present the top eight most active fractions, which exhibit MMP activities ranging from 60% to 133%. Additionally, we detail their CyQuant, MTT, and MMP activities, along with the corresponding mitoprotective compounds identified from these fractions.

These fractions were selected for orbitrap LCMS chemical profiling, and the data were analyzed using compound discoverer 3.3 with a native untargeted natural products identification workflow. Mass data were searched in the Chemspider database, while a spectral similarity search was performed in mZcloud for MS^2^ fragmentation data of detected compounds. Compounds with at least 70% mZcloud match confidence and a mass error between –2 ppm and +2 ppm were selected, resulting in the identification of 57 compounds across eight fractions. Table 4 shows a list of identified compounds from all the positive biota fractions with their retention times, molecular formulae, and molecular weights. The extracted ion chromatogram and mass spectra of analyzed fractions are included in the Appendix A.

It is noteworthy that 12 compounds identified by LC-MS have been reported to protect against mitochondrial dysfunction, underscoring the robustness of our screening methodology (Figure 8). Research has shown that the majority of these compounds protect the mitochondria by reducing oxidative stress and modulating antioxidant defense systems, oxidative stress, and apoptotic markers [45,68,69]. In the SH-SY5Y model, fraxetin (**1**) protected against rotenone-induced apoptosis via the induction of the chaperone HSP70, crucial for maintaining mitochondrial functions [70]. Fraxetin also protects mitochondria by reducing ROS and apoptotic proteins, such as cytochrome c, Bax, and caspase-3 and -9, while upregulating the expression of antioxidant defense enzymes, such as SOD, GPX, and catalase [69,70]. Nicotinamide (**2**), a precursor to nicotinamide adenine dinucleotide (NAD), has been reported to prevent neurodegeneration in glaucoma by defending against mitochondrial dysfunction, boosting OXPHOS, and increasing mitochondrial size [68]. In addition to reducing ROS, regulating oxidative enzymes, and increasing the expression of complexes I and V, naringenin (**3**) upregulates ATP production, activates the PI3K/Akt/GSK-3β pathway, improves nuclear E2-related factor 2 (Nrf2) expression, and stabilizes MMP [45,71,72,73]. Isoliquiritigenin (**4**) inhibits apoptosis by promoting increased phosphorylation of glycogen synthase kinase-3β (GSK3β) and also enhances mitochondrial biogenesis by activation of AMP-activated protein kinase (AMPK) [74,75]. Reports have shown that pinocembrin (**5**) protects brain mitochondria structure and function by decreasing ROS, restoring MMP, and improving mitochondrial morphology [76]. This compound also exhibits antiapoptotic effects, restores the electron transport chain, and upregulates ATP and Nrf2 [77,78]. Other compounds like quercetin (**6**) [30,31,79], taxifolin (**7**) [80,81,82], agmatine (**8**) [83], esculin (**9**) [84], vanillin (**10**) [85], syringic acid (**11**) [86,87,88], and 4-coumaric acid (**12**) [43,89] have also been documented for their protective effects against mitochondrial dysfunction.

We also identified 45 new mitoprotective compounds spanning diverse structural classes, such as alkaloids, coumarins, carboxylic acids, cinnamic acids, lipopeptides, terpenes, benzothiazoles, amines, amino acids, and fatty acids. While there are no reports in the literature on the mitoprotective activity of these metabolites, some of them have been reported to show activity in mitochondrial dysfunction-linked diseases. For example, trigonelline, an alkaloid previously isolated from plants such as fenugreek, Japanese radish, and coffee beans stands out for its potential to attenuate oxidative stress and show activity in mitochondrial dysfunction-linked conditions, such as diabetes, PD, AD, stroke, dementia, and depression [90]. Trigonelline improved memory function in AD and showed neuroprotective and antiapoptotic effects in a 6-OHDA-induced model of PD in rats [91,92,93]. Our work demonstrated that these beneficial effects may at least partially be attributed to its mitoprotective functions. Isoferulic acid and scopoletin exert their neuroprotective properties by decreasing ROS, activating the Nrf2 pathway, and suppressing apoptosis [94,95,96,97]. In this study, they were found to be promising candidates for the mitoprotective effects of *Alnus* sp. and *Ternstroemia* sp., respectively (Table 4).

## 4. Conclusions, Limitations, and Future Directions

In this study, we screened 4224 fractions derived from 384 biotas belonging to Australia’s diverse flora and fauna for their protective effects against mitochondrial dysfunction. We were able to identify 20 hit fractions from 11 biota in the initial round of screening using assays based on rotenone-induced mitochondrial dysfunction in PD patient-derived hONS cell models. These initial findings were successfully validated by MTT and MMP assays in a 6-OHDA/SH-SY5Y model. Additionally, we identified 57 metabolites that are potentially responsible for the activity of the eight most active hits using HRMS. Twelve metabolites (**1**–**12**) have been previously reported to show protective effects against mitochondrial dysfunction.

These findings strongly indicate that the described methodologies provide a robust, effective, and rather quick approach to the screening of mitochondrial modulators, especially on a large scale. With the use of robotics, the screening approach described in this work can be easily scaled to screen tens of thousands of compounds. Also, this method can be readily adapted to screen for toxic mitochondrial modulators as a strategy for cancer therapy, rather than for mitoprotective compounds, which was the focus of this work.

However, a significant limitation of this work is the reliance on databases for identifying compounds, the results being inherently dependent on the contents of these databases. Consequently, there is a considerable risk that both known and new natural products have been overlooked. This challenge underscores the importance of further isolation and characterization of active compounds from these biotas to precisely determine the metabolites responsible for their mitoprotective properties and further testing of these metabolites to understand their mechanisms of actions.

## Figures and Tables

**Figure 1 biomolecules-14-00440-f001:**
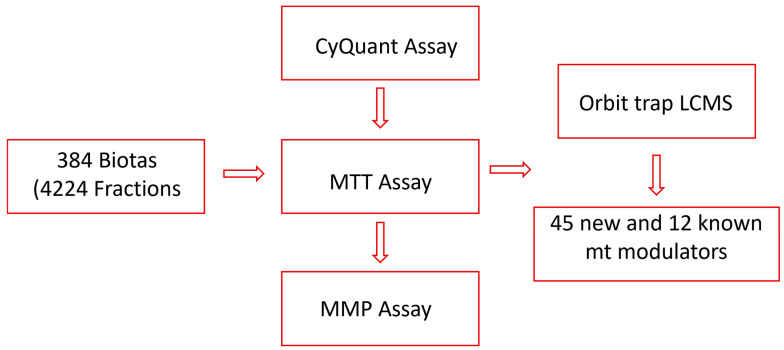
Flow chart of the study.

**Figure 2 biomolecules-14-00440-f002:**
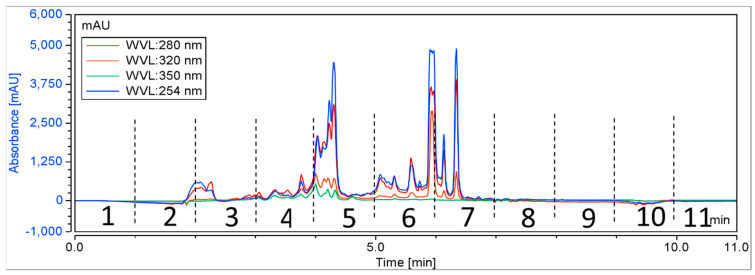
A schematic chromatogram illustrating the HPLC fractionation of a representative positive biota. Dotted lines indicate the time periods for each fraction.

**Figure 3 biomolecules-14-00440-f003:**
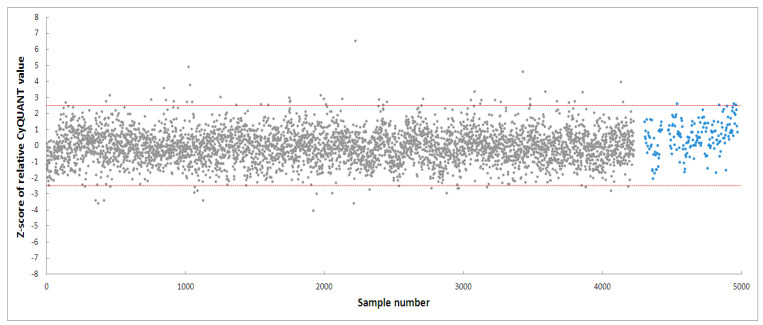
High-throughput screening of 4224 fractions obtained from NatureBank generating 108 hits. The two red lines indicate a relative Z-score of −2.5–2.5. Grey dots: fractions, blue dots: cells treated with vehicle only.

**Figure 4 biomolecules-14-00440-f004:**
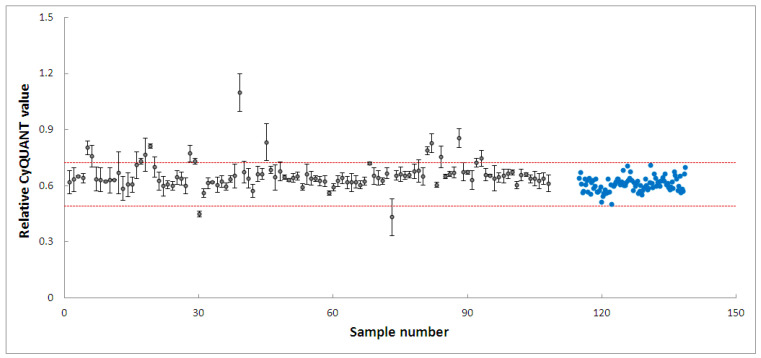
High-throughput screening of 108 fractions, leading to 20 fractions from 11 biotas. Grey dots: fractions, blue dots: cells treated with vehicle only. The two red lines indicate a relative Z-score of −2.5–2.5.

**Figure 5 biomolecules-14-00440-f005:**
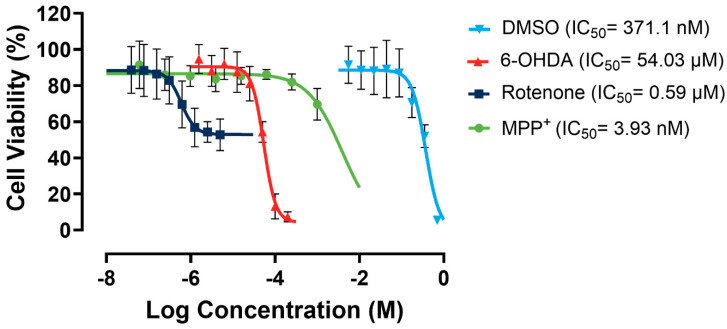
Dose–response curves for rotenone, MPP^+^, 6-OHDA, and DMSO in SH-SY5Y cells. Assays were performed in three biological repeats, with two technical repeats for each biological repeat.

**Figure 6 biomolecules-14-00440-f006:**
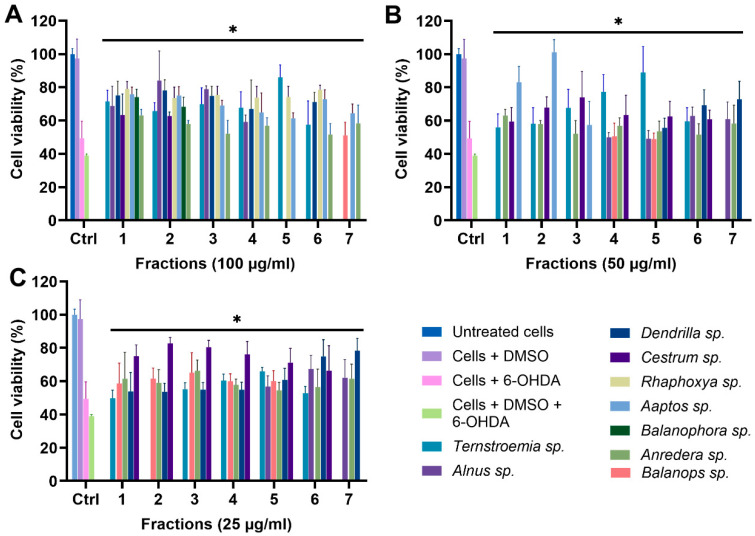
MTT assay of fractions at (**A**) 100 µg/mL, (**B**) 50 µg/mL, and (**C**) 25 µg/mL. * *p* < 0.05 when fractions are compared to cells + DMSO + 6-OHDA.

**Figure 7 biomolecules-14-00440-f007:**
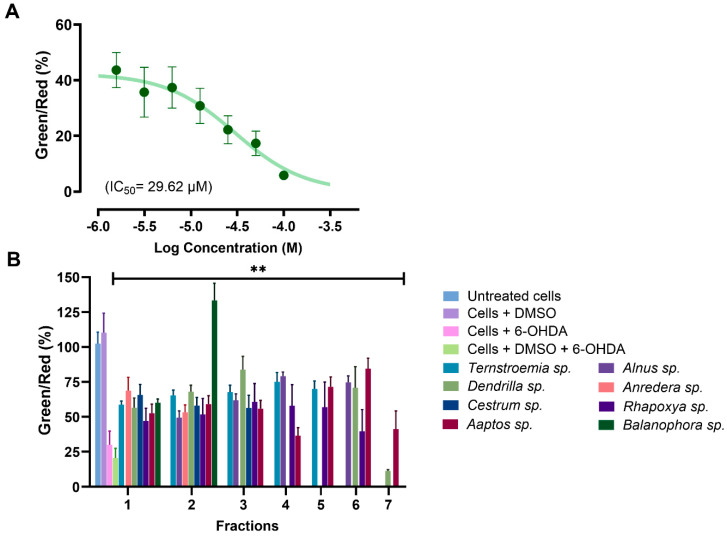
(**A**) Dose–response curve for SH-SY5Y cells exposed to 6-OHDA using the mitochondrial membrane potential (MMP) assay. (**B**) Analysis of selected active fractions identified in the MTT assay through the MMP assay. All the analyzed fractions, except for fraction 7 from *Dendrilla* sp., showed significant mitochondrial protective effects compared to the control group of cells treated with DMSO and 6-OHDA (** *p* < 0.01).

**Figure 8 biomolecules-14-00440-f008:**
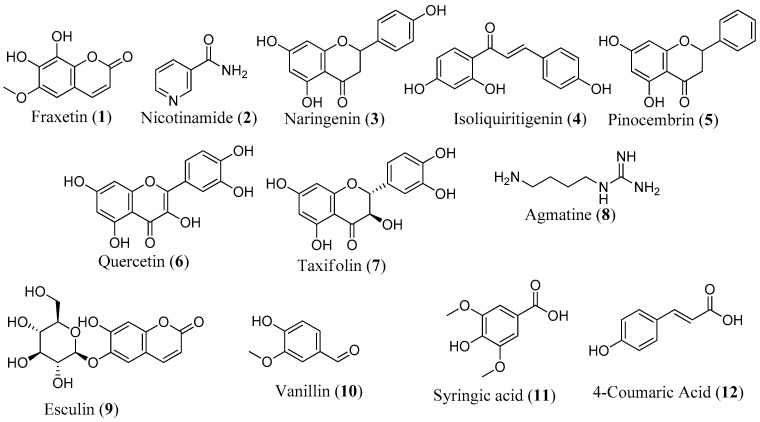
Structure of compounds with known mitoprotective activity.

**Table 1 biomolecules-14-00440-t001:** HPLC solvent gradient.

Time	MeOH (%)	H_2_O (%)	Flow Rate (mL)
0.01	10	90	4
3.00	50	50	4
3.01	50	50	3
6.50	100	0	3
7.00	100	0	3
7.00	100	0	4
8.00	100	0	4
9.00	10	90	4
11.00	10	90	4

**Table 2 biomolecules-14-00440-t002:** List of eleven biotas from which hits were obtained during the second round of screening.

	Genus	Species	Source
1	*Balanops*	<None>	Plant
2	*Alnus*	*trabeculosa h-m.*	Plant
3	*Anredera*	<None>	Plant
4	*Furcraea*	<None>	Plant
5	*Cestrum*	<None>	Plant
6	*Ilex*	<None>	Plant
7	*Balanophora*	<None>	Plant
8	*Rhaphoxya*	3249	Marine
9	*Aaptos*	*aaptos*	Marine
10	*Dendrilla*	3106	Marine
11	*Ternstroemia*	<None>	Plant

**Table 3 biomolecules-14-00440-t003:** Summary data for all eight fractions and identified mitoprotective compounds.

	CyQuant	MTT (%)	MMP (%)	Compound(s)
**Cell + 6-OHDA + DMSO**	✕	38.98	20.58	-
***Ternstroemia*** **sp._F4**	√	67.70	75.04	1, 4, 6, 7, 9, 10, 12
***Cestrum*** **sp._F1**	√	63.39	65.69	4, 5, 8,
***Alnus*** **sp._F4**	√	59.24	79.15	1, 3, 10, 11
***Balanophora*** **sp._F2**	√	68.27	133.39	2
***Anredera*** **sp._F1**	√	65.82	68.76	10
***Aaptos*** **sp._F6**	√	72.83	84.52	-
***Dendrilla*** **sp._F3**	√	74.82	83.90	8, 10
***Rhaphoxya*** **sp._F3**	√	75.44	60.78	-

**Table 4 biomolecules-14-00440-t004:** Identified compounds from UHPLC-Orbitrap Ms.

	Name	RT (min)	MF	MW
***Ternstroemia*** **sp._F4**				
1	NP-019811	0.56	C_6_H_7_NO_2_	125.04765
2	5,7-Dihydroxy-2-(4-hydroxyphenyl)-6,8-bis [3,4,5-trihydroxy-6-(hydroxymethyl)tetrahydro-2H-pyran-2-yl]-4H-chromen-4-one	0.97	C_27_H_30_O_15_	594.15819
3	(2S,3R,4S,5S,6R)-2-[4-(2-hydroxyethyl)phenoxy]-6-(hydroxymethyl)oxane-3,4,5-triol	1.06	C_14_H_20_O_7_	300.12041
4	Esculin	1.07	C_15_H_16_O_9_	340.07921
5	7-hydroxy-6-methoxy-2H-chromen-2-one	1.15	C_10_H_8_O_4_	192.04201
6	Taxifolin	1.37	C_15_H_12_O_7_	304.05798
7	Fraxetin	1.38	C_10_H_8_O_5_	208.03696
8	3,4-Dihydroxybenzaldehyde	1.42	C_7_H_6_O_3_	138.03163
9	5-(6-hydroxy-6-methyloctyl)-2,5-dihydrofuran-2-one	2.26	C_13_H_22_O_3_	226.15672
10	Vanillin	2.45	C_8_H_8_O_3_	152.04727
11	(2R,3S,4S,5R,6R)-2-(hydroxymethyl)-6-(2-phenylethoxy)oxane-3,4,5-triol	2.96	C_14_H_20_O_6_	284.12577
12	Scopoletin	2.96	C_10_H_8_O_4_	192.04216
13	4-Coumaric acid	3.00	C_9_H_8_O_3_	164.04738
14	Isoliquiritigenin	3.30	C_15_H_12_O_4_	256.07338
15	Quercetin	3.88	C_15_H_10_O_7_	302.04265
16	5-hydroxy-3-(4-methoxyphenyl)-7-[(3,4,5-trihydroxy-6-{[(3,4,5-trihydroxy-6-methyloxan-2-yl)oxy]methyl}oxan-2-6-methyloxan-2-yl)oxy]methyl}oxan-2-yl)oxy]-4H-chromen-4-one	4.03	C_28_H_32_O_14_	592.17931
17	3-amino-2-phenyl-2H-pyrazolo [4,3-c]pyridine-4,6-diol	5.48	C_12_H_10_N_4_O_2_	242.08023
18	(3aR,7aS,8S,9aR)-5,8-dimethyl-3-methylidene-2H,3H,3aH,4H,6H,7H,7aH,8H,9H,9aH-azuleno [6,5-b]furan-2,6-dione	7.16	C_15_H_18_O_3_	246.12546
19	12-Aminododecanoic acid	8.90	C_12_H_25_NO_2_	215.18846
***Cestrum*** **sp._F1**				
1	Agmatine	0.45	C_5_H_14_N_4_	130.12168
2	Nicotinic acid	0.83	C_6_H_5_NO_2_	123.03203
3	N,N′-Diphenylguanidine	0.92	C_13_H_13_N_3_	211.11072
4	Vanillin	2.44	C_8_H_8_O_3_	152.04753
5	NP-007065	4.04	C_8_H_10_O_3_	154.06290
6	4-Phenylbutyric acid	4.27	C_10_H_12_O_2_	164.08365
7	Cantharidin	9.05	C_10_H_12_O_4_	196.07384
8	NP-000925	9.87	C_17_H_16_O_5_	300.09947
9	Pinocembrin	10.06	C_15_H_12_O_4_	256.07314
***Alnus*** **sp._F4**				
1	NP-019811	0.55	C_6_H_7_NO_2_	125.04747
2	Nicotinic acid	0.87	C_6_H_5_NO_2_	123.03188
3	2,3,4,9-Tetrahydro-1H-β-carboline-3-carboxylic acid	1.35	C_12_H_12_N_2_O_2_	216.08965
4	Fraxetin	2.15	C_10_H_8_O_5_	208.03697
5	Syringic acid	3.13	C_9_H_10_O_5_	198.05271
6	Isovanillic acid	3.33	C_8_H_8_O_4_	168.04217
7	Isoferulic acid	3.33	C_10_H_10_O_4_	194.05776
8	Naringenin	7.06	C_15_H_12_O_5_	272.06819
9	Vanillin	8.01	C_8_H_8_O_3_	152.04720
***Balanophora*** **sp._F2**				
1	Nicotinamide	0.78	C_6_H_6_N_2_O	122.04791
2	3′-Adenosine monophosphate (3′-AMP)	0.98	C_10_H_14_N_5_O_7_P	347.06261
3	3-(2-methylpropyl)-octahydropyrrolo [1,2-a]pyrazine-1,4-dione	3.03	C_11_H_18_N_2_O_2_	210.13658
4	NP-007065	4.00	C_8_H_10_O_3_	154.06297
***Anredera*** **sp._F1**				
1	L-Aspartic acid	0.66	C_4_H_7_NO_4_	133.03739
2	Nicotinic acid	0.86	C_6_H_5_NO_2_	123.03193
3	L-Phenylalanine	0.88	C_9_H_11_NO_2_	165.07886
4	Vanillin	2.43	C_8_H_8_O_3_	152.04729
5	4-Phenylbutyric acid	3.13	C_10_H_12_O_2_	164.08362
***Aaptos*** **sp._F6**				
1	Pulegone	7.95	C_10_H_16_O	152.12012
2	NP-022512	8.67	C_13_H_19_NO	205.14656
3	NP-019636	15.25	C_9_H_8_O_4_	180.04219
4	Palmitoleic acid	16.69	C_16_H_30_O_2_	254.22444
5	4-Methoxycinnamic acid	17.15	C_10_H_10_O_3_	178.06291
6	8Z,11Z,14Z-Eicosatrienoic acid	17.66	C_20_H_34_O_2_	306.25477
***Dendrilla*** **sp._F3**				
1	Agmatine	0.49	C_5_H_14_N_4_	130.12166
2	Trigonelline	0.82	C_7_H_7_NO_2_	137.04752
3	Nicotinic acid	0.82	C_6_H_5_NO_2_	123.03189
4	Vanillin	2.41	C_8_H_8_O_3_	152.04744
5	NP-011220	2.77	C_11_H_18_N_2_O_2_	210.13654
6	4-Phenylbutyric acid	4.24	C_10_H_12_O_2_	164.08368
7	Cantharidin	9.04	C_10_H_12_O_4_	196.07358
8	(-)-Caryophyllene oxide	11.31	C_15_H_24_O	220.18263
9	4-Phenylbutyric acid	11.76	C_10_H_12_O_2_	164.08377
***Rhaphoxya*** **sp._F3**				
1	Nicotinic acid	0.82	C_6_H_5_NO_2_	123.03196
2	3-[(4-hydroxyphenyl)methyl]-octahydropyrrolo [1,2-a]pyrazine-1,4-dione	1.24	C_14_H_16_N_2_O_3_	260.11604
3	NP-016455	2.39	C_11_H_18_N_2_O_4_	242.12662
4	NP-011220	2.77	C_11_H_18_N_2_O_2_	210.13680
5	Cyclo(leucylprolyl)	3.01	C_11_H_18_N_2_O_2_	210.13684
6	DL-2-(acetylamino)-3-phenylpropanoic acid	3.37	C_11_H_13_NO_3_	207.08954
7	4-Phenylbutyric acid	3.38	C_10_H_12_O_2_	164.08382
8	Cyclo(phenylalanyl-prolyl)	3.70	C_14_H_16_N_2_O_2_	244.12128
9	4-Methylumbelliferone hydrate	4.48	C_10_H_8_O_3_	176.04732
10	2-Hydroxybenzothiazole	5.03	C_7_H_5_NOS	151.00932
11	4-Methoxycinnamaldehyde	7.03	C_10_H_10_O_2_	162.06818
12	Bis(2-ethylhexyl) amine	8.66	C_16_H_35_N	241.27669
13	(1R,4aS)-7-(2-Hydroxypropan-2-yl)-1,4a-dimethyl-9-oxo-3,4,10,10a-carboxylic acid	10.63	C_20_H_26_O_4_	330.18295
14	(-)-Caryophyllene oxide	12.54	C_15_H_24_O	220.18265

## Data Availability

Data are available in the manuscript text or in the Appendix A.

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
