# Peer review of "A High-Throughput Screening of a Natural Products Library for Mitochondria Modulators"

_biomolecules, 2024, doi:10.3390/biom14040440_

Round 1

Reviewer 1 Report

Comments and Suggestions for Authors

1. The manuscript entitled “A high throughput screening of natural products library for mitochondria modulators” authored by Emmanuel Makinde et al offers valuable insights into natural products as mitochondria modulators. The main question addressed by the research refers to the screening of selected natural products - Australian plants and marine sponges- to develop mitochondria modulators.

2. Overall, I found the manuscript to be scientifically sound, with data relevant to the field, still the novelty and originality are poorly presented by the authors and these aspects should be addressed by the authors for abstract, introduction and conclusion sections.

The research addressed a particular methodology for the preliminary screening - PD patient-sourced olfactory neurosphere-derived (hONS) cells with rotenone, followed by the actual testing of selected biota samples’ mitochondria modulatory activities on human neuroblastoma (SH-SY5Y) cells using a MTT assay targeting mitochondrial nicotinamide adenine dinucleotide phosphate (NADPH)-dependent dehydrogenases and a mitochondrial membrane potential (MMP) assay.

The authors provide a comprehensive analysis of the biota samples, with suitable and relevant protocols.

Still, several sections of the manuscript could be improved:

1.     Introduction – too concise, with vague presentation of the link between mitochondria impairment and neurodegenerative diseases. Lines 36-42 are not sufficient if the authors intended to develop the information from the abstract introduction. The authors should provide a detailed presentation of the link between mitochondria impairment and neurodegenerative diseases, as well as the of mitochondria modulators potential. Previous studies should be added to underline the novelty of the research – a concise paragraph for the introduction section, a more detailed one for the discussion section

The vocabulary for the whole introduction section needs to be improved – many words repetitions, in particular, line 36-42

3. As mentioned above, the authors are recommended to underline the novelty of the research – a presentation of previous studies and a distinct paragraph presenting the novelty of their research

4. The methodology is comprehensively described, still the authors should add the literature reference(s) for all protocols used.

5.  The conclusions section of the current form of the manuscript does not have contain conclusions, it lists the results. The authors are recommended to underline the novelty. As part of the discussion sections, the authors should address the study limitations.

6. Several references are dated, the authors should add references connected to the similar studies used to enrich the introduction and discussion section

7. Additional comments on the tables and figures and quality of the data.

The manuscript does not follow the MDPI template

Author Response

Reviewer 1

  1.    Introduction – too concise, with vague presentation of the link between mitochondria impairment and neurodegenerative diseases. Lines 36-42 are not sufficient if the authors intended to develop the information from the abstract introduction. The authors should provide a detailed presentation of the link between mitochondria impairment and neurodegenerative diseases, as well as the of mitochondria modulators potential.

The vocabulary for the whole introduction section needs to be improved – many words repetitions, in particular, line 36-42

 Response:

We thank the reviewer for this comment. To demonstrate a link between mitochondria impairment and neurodegenerative diseases, we have rewritten paragraph 2 and included a new paragraph 3

Line 35-40

“Neurons, with their high energy demands, contain hundreds of thousands of mitochondria, which are responsible to meet most of their ATP needs for functioning of the CNS [6,11,12]. The quality of these mitochondria is critical, as they must be highly functional to support the complex activities of neurons in the CNS [6]. Dysfunction in mitochondria, particularly in the oxidative phosphorylation (OXPHOS) system, has been linked to various diseases, notably neurodegenerative disorders like Parkinson’s disease (PD), Huntington’s disease, and Alzheimer’s disease [2,4,6–9]. The mechanisms behind these links are multifaceted.”

Line 41-57

“Firstly, diminished ATP production due to impaired mitochondrial function plays a significant role in the energy deficits observed in neurons affected by these diseases. Such deficits can compromise neuronal function and survival, potentially leading to cell death [3,5,10]. Secondly, about 90% of ROS are generated as by-products of the OXPHOS process, about 90% of ROS are generated as by-products of the OXPHOS process [1,6,11,12]. Although ROS serve as signaling molecules under normal conditions, their overproduction or the failure of antioxidant defenses can induce oxidative stress, harming DNA, proteins, and lipids. This oxidative damage, a common feature of neurodegenerative diseases, likely drives further neuronal damage [3,5]. Thirdly, mitochondria play a key role in regulating intracellular calcium levels, which are crucial for various cellular processes, including neurotransmitter release, synaptic plasticity, and cell survival. Mitochondrial dysfunction can lead to dysregulated calcium homeostasis, exacerbating neuronal injury and death [1,6]. Fourthly, impaired mitophagy, a specific form of autophagy that removes damaged mitochondria from the cell, has been linked to neurodegenerative diseases such as PD [13,14]. Mutations in genes like PINK1 and Parkin, which are involved in mitophagy, can cause genetic PD, underscoring the importance of mitochondrial quality control in neurodegeneration [15,16]. Lastly, mitochondria play a crucial role in the intrinsic apoptosis pathway by releasing pro-apoptotic factors such as cytochrome C. Dysregulation of apoptotic signaling pathways due to mitochondrial dysfunction can trigger inappropriate neuronal cell death, contributing to neurodegeneration [12,17,18].”

In the discussion we have also rewritten paragraph 1 of section 3.2 as in line 192-197.

“The effect of fractions used in this study on cell proliferation was evaluated by exposing PD-sourced hONS cells to rotenone in the presence of test fractions. There is a well-established link between PD and mitochondrial dysfunction as extensive research evidence have linked PD to features of mitochondrial dysfunction such as oxidative stress, poor calcium homeostasis, disruption to the MMP, abnormal mitochondrial morphology, disruptions to the OXPHOS process as a result of reduced complex enzyme activity, imbalanced apoptosis, impaired mitophagy and mitochondrial dynamics [6,16,34–37].”

  1. Previous studies should be added to underline the novelty of the research – a concise paragraph for the introduction section, a more detailed one for the discussion section

Response:

We have added a new paragraph in line 58-68

“Recognizing the pivotal role of mitochondrial function in various diseases, extensive research has focused on identifying compounds that can modify mitochondrial functions. Compounds such as berberine, resveratrol, and epigallocatechin-3-gallate are known to trigger mitochondrial biogenesis and influence mitochondrial dynamics by promoting both fusion and fission [19–25]. Curcumin and its derivatives have been demonstrated to regulate mitochondrial dynamics to remedy dysfunction, and flavonoids like quercetin have shown potential in ameliorating memory impairment through mitochondrial regulation [26–31]. Additionally, compounds such as polydatin and acacetin have been found to induce mitophagy, enhancing mitochondrial function and offering protective effects in disease models [32,33]​. Although these products show promise in enhancing mitochondrial function and offering disease model protection, the variability in the effectiveness of these compounds across different cell types and disease models highlights the need for more targeted approaches in their application and calls for novel, safe and more potent mitochondrial modifiers”.

  1. As mentioned above, the authors are recommended to underline the novelty of the research – a presentation of previous studies and a distinct paragraph presenting the novelty of their research.

Response:

To highlight the novelty of our research, we have expanded on the distinctive approach employed in our study. Furthermore, we have added an in-depth discussion on the compounds that exhibit mitochondrial protective effects newly identified in this work, underscoring their potential significance in advancing our understanding of mitochondrial function and protection.

Paragraph 6, line 80-95

“A common approach utilized to search for mitoprotective metabolites is to test compounds largely based on previously reported antioxidant or neuroprotective activity [43–45]. This work diverges from that by screening a large library of natural products to boost the chances of finding novel mitoprotective compounds. Overall, we have successfully established and implemented a robust process for the identification of mitoprotective compounds from natural products using a stepwise combination of three different assays, two cell lines and two mitochondrial toxins. We have also described how each assay seamlessly dovetails into the next, leading to the identification of 57 metabolites including 45 new mitoprotective compounds.”

We have also included a more detailed paragraph in the discussion section, line 346-357

“We also identified 45 new mitoprotective compounds, spanning diverse structural classes such as alkaloids, coumarins, carboxylic acids, cinnamic acids, lipopeptides, terpenes, benzothiazoles, amines, amino acids and fatty acids. While there are no reports in literature on the mitoprotective activity of these metabolites, some of them have been reported to show activity in mitochondrial dysfunction linked diseases. For example, trigonelline, an alkaloid previously isolated from plants such as fenugreek, Japanese radish and coffee beans, stands out for its potential to attenuate oxidative stress and show activity in mitochondrial dysfunction linked conditions such as diabetes, PD, AD, stroke, dementia and depression [89]. Trigonelline improved memory function in AD and showed neuroprotective and antiapoptotic effect in 6-OHDA induced model of PD in rats [90–92]. Our work demonstrated that these beneficial effects may at least partially be attributed to its mitoprotective functions. Isoferulic acid and scopoletin exerts its neuroprotective properties by decreasing ROS, activation of Nrf2 pathway and suppression of apoptosis [93–96]. In this study, they are promising candidates for the mitoprotective effects of Alnus sp. and Ternstroemia sp. respectively (Table 4).”

  1. The methodology is comprehensively described, still the authors should add the literature reference(s) for all protocols used.

Response:

Our methodologies were largely based on established protocols from manufacturers. This have been expressly stated in:

Line 131-132

“CyQUANT Cell Proliferation Assay Kit (Life Technologies) was used to evaluate the cellular response of hONS cells to rotenone as described by the manufacturer.”

Line 139

“MTT was obtained from Sigma and the assay was performed as described by the manufacturer…”

Line 149-151

“Codex Homogeneous Mitochondrial Membrane Potential Assay Kit (Codex BioSolutions, Inc., Gaithersburg, MD) was used for the MMP assay, the assay was performed as described by manufacturer with slight modifications.”

  1. The conclusions section of the current form of the manuscript does not have contain conclusions, it lists the results. The authors are recommended to underline the novelty. As part of the discussion sections, the authors should address the study limitations.

 Response:

The conclusion has been improved as suggested by the reviewer we have also mentioned the limitation of the study as part of the conclusion.

In Line  366-377

“These findings strongly indicate that the described methodologies provide a robust, effective and rather quick approach to the screening of mitochondrial modulators especially on a large scale. With the use of robotics, the screening approach described in this work can be easily scaled to screen tens of thousands of compounds. Also, this method can be readily adapted to screen for toxic mitochondrial modulators as a strategy for cancer therapy, rather than mitoprotective compounds that are the focus of this work.

However, a significant limitation of this work is the reliance on databases for identifying compounds, which are inherently dependent on the contents of these databases. Consequently, there is a considerable risk that both known and new natural products have been overlooked. This challenge underscores the importance of further isolation and characterization of active compounds from these biotas to precisely determine the metabolites responsible for their mitoprotective properties and further testing of these metabolites to understand their mechanisms of actions.” 

  1. Several references are dated, the authors should add references connected to the similar studies used to enrich the introduction and discussion section.

Response:

The authors have been careful to ensure that majority of the references cited are within the last 5 to 10 years except in a few cases. In situations where an older article is cited, they’re usually accompanied by a more recent citation except in cases where we couldn’t find one.

For example, the introduction contains references 1-45

Reference 27 (2013) cited alongside 28 (2018), 34 and 36 (2013) with 35 (2020), 41 (2021) with 42 (2012), 39 was the best and most detailed on CyQuant assay we could find.

The discussion contains references 36-96

58 (1995) cited with 35 (2021)

68 and 69 (both 2005) are the only references for fraxetin as a mitoprotective compound.

73 (2010) for isoliquiritigenin

75 (2006) for pinocembrin

84 (2000) for vanillin

  1. Additional comments on the tables and figures and quality of the data.

 Response:

We have added additional information to explain the content of Tables 3 and 4.

Line 309-311

In Table 3, we present the top eight most active fractions, which exhibit MMP activities ranging from 60% to 133%. thAdditionally, we detail their CyQuant, MTT and MMP activity, along with the corresponding mitoprotective compounds identified from these fractions.

Line 317-318

Table 4 shows a list of identified compounds from all the positive biota fractions, their retention time, molecular formular and molecular weights.

  1. The manuscript does not follow the MDPI template

Response:

Manuscript was submitted without formatting because Biomolecules accepts free format submission. All formatting will be done as required.

Reviewer 2 Report

Comments and Suggestions for Authors

In this manuscript, the authors developed a high throughput method to screen 4224 fractions protective effect against mitochondrial dysfunction.  Their robust and meticulous approach led to the identification of 20 hit fractions 11 biota in the first round of screening using CyQuant cell viability assay on PD patient-derived hONS cells.  They successfully validated this by MTT and MMP assays on a neuroblastoma cell line.  The authors discovered 20 hit fractions and out of the 8 most active hits, they found 57 metabolites that are potentially responsible for the bioactivity using HRMS.  In general, the manuscript is very well written and the conclusions drawn were well supported by the results provided.  The high throughput method and strategy described in this manuscript should serve as a template for others to find metabolites from other natural products.

Major comments

The authors mentioned that they have identified 57 metabolites using HRMS that are potentially responsible for the activity of 8 most active hits. Out of these, 12 metabolites have been previously shown to have protective effect against mitochondrial dysfunction.  What about the remaining 45 metabolites?  Perhaps, a short discussion on these 45 metabolites would be helpful. E.g., are there potential new compounds amongst the 45 metabolites.

Minor comments

Typo on line 18, ‘effecacy’ should read ‘efficacy’.

Typo on line 129, ‘Phenomonex’ should read ‘Phenomenex’.

Author Response

The authors thank the reviewers for their comments. The comments have been carefully considered and addressed as advised.

Reviewer 2

  1. The authors mentioned that they have identified 57 metabolites using HRMS that are potentially responsible for the activity of 8 most active hits. Out of these, 12 metabolites have been previously shown to have protective effect against mitochondrial dysfunction.  What about the remaining 45 metabolites?  Perhaps, a short discussion on these 45 metabolites would be helpful. E.g., are there potential new compounds amongst the 45 metabolites.

Response:

We thank the reviewer for this comment. We have addressed the 45 metabolites in line 346-357.

“We also identified 45 new mitoprotective compounds, spanning diverse structural classes such as alkaloids, coumarins, carboxylic acids, cinnamic acids, lipopeptides, terpenes, benzothiazoles, amines, amino acids and fatty acids. While there are no reports in literature on the mitoprotective activity of these metabolites, some of them have been reported to show activity in mitochondrial dysfunction linked diseases. For example, trigonelline, an alkaloid previously isolated from plants such as fenugreek, Japanese radish and coffee beans, stands out for its potential to attenuate oxidative stress and show activity in mitochondrial dysfunction linked conditions such as diabetes, PD, AD, stroke, dementia and depression [89]. Trigonelline improved memory function in AD and showed neuroprotective and antiapoptotic effect in 6-OHDA induced model of PD in rats [90–92]. Our work demonstrated that these beneficial effects may at least partially be attributed to its mitoprotective functions. Isoferulic acid and scopoletin exerts its neuroprotective properties by decreasing ROS, activation of Nrf2 pathway and suppression of apoptosis [93–96]. In this study, they are promising candidates for the mitoprotective effects of Alnus sp. and Ternstroemia sp. respectively (Table 4).”

  1. Minor comments

Typo on line 18, ‘effecacy’ should read ‘efficacy’.

Typo on line 129, ‘Phenomonex’ should read ‘Phenomenex’.

Response:

‘effecacy’ corrected as ‘efficacy’ in line 18.

‘Phenomonex’ corrected as ‘Phenomenex’ in line 165.

Reviewer 3 Report

Comments and Suggestions for Authors

The authors have conducted an extensive screening of 4224 fractions from 384 Australian marine organisms and plant samples to identify natural products with protective effects on the mitochondria. It is such a huge time-consuming work. I think the work is ready to go if they can make minor revisions: 

1) since it is a screening work, a flow chart should be incorporated;

2) is there any possible to validate their screening assay? how reliable the assay is?

Author Response

The authors thank the reviewers for their comments. The comments have been carefully considered and addressed as advised.

Reviewer 3

1) since it is a screening work, a flow chart should be incorporated;

Response: the authors thank the reviewer for this comment, a flow chart has been incorporated as Figure 1 in line 87. Figure legends have been updated accordingly.

Figure 1. Flow chart of the study.

  1. 2. Is there any possible to validate their screening assay? how reliable the assay is?

Response:

The assays were validated by positive controls – the results of these positive controls were comparable with those reported in the literature [1–3].

We consider our assays to be reliable. Assays were repeated 9 times as 3 technical repeats of 3 separate biological repeats or experiments.

As discussed in the 3.5 and 3.6 data from the first round of screening using CyQuant assay were corroborated by the second round of screening using MTT assay. This was later confirmed using MMP assay where 33 out of 34 tested fractions returned positive. In addition, we successfully identified 12 compounds that have been previously shown to have mitoprotective effects, which strongly supports the high efficiency and reliability of our strategy.

References

  1. Wang, S.-F.; Liu, L.-F.; Wu, M.-Y.; Cai, C.-Z.; Su, H.; Tan, J.; Lu, J.-H.; Li, M. Baicalein Prevents 6-OHDA/Ascorbic Acid-Induced Calcium-Dependent Dopaminergic Neuronal Cell Death. Sci Rep 2017, 7, 8398, doi:10.1038/s41598-017-07142-7.
  2. Rehfeldt, S.C.H.; Laufer, S.; Goettert, M.I. A Highly Selective In Vitro JNK3 Inhibitor, FMU200, Restores Mitochondrial Membrane Potential and Reduces Oxidative Stress and Apoptosis in SH-SY5Y Cells. International Journal of Molecular Sciences 2021, 22, 3701, doi:10.3390/ijms22073701.
  3. Kesh, S.; Kannan, R.R.; Balakrishnan, A. Naringenin Alleviates 6-Hydroxydopamine Induced Parkinsonism in SHSY5Y Cells and Zebrafish Model. Comparative Biochemistry and Physiology Part C: Toxicology & Pharmacology 2021, 239, 108893, doi:10.1016/j.cbpc.2020.108893.

Round 2

Reviewer 1 Report

Comments and Suggestions for Authors

The authors took into consideration and performed all the suggested modifications

One observation  - I underline the need to include references for all used protocols and to state that these protocols are standardized 

  1. The methodology is comprehensively described, still the authors should add the literature reference(s) for all protocols used.

Author Response

The authors thank the reviewer for the comment. The comment has been carefully considered and addressed as advised.  All changes in the manuscript in this round of review are highlighted in blue

Reviewer 1

  1. The methodology is comprehensively described, still the authors should add the literature reference(s) for all protocols used.

Response:

Reference has been added for CyQuant, MTT and MMP assay in lines 131-132, 139 and 148-150 respectively.

Line 131-132

“CyQUANT Cell Proliferation Assay Kit (Life Technologies) was used to evaluate the cellular response of hONS cells to rotenone as described by Murtaza et al. [47].”

Line 139“MTT assay was performed as previously described [51].”

Line 149-150

“Codex Homogeneous Mitochondrial Membrane Potential Assay Kit (Codex BioSolutions, Inc., Gaithersburg, MD) was used for the MMP assay, the assay was performed in line with the manufacturer’s protocol and as described by Sakamuru et al. [52] with some modifications.”